# Ambient air pollution and urological cancer risk: A systematic review and meta-analysis of epidemiological evidence

Jinhui Li [1,7] ✉, Zhengyi Deng [1,7], Simon John Christoph Soerensen [1,2], Linda Kachuri [2,3], Andres Cardenas [2], Rebecca E. Graff [4], John T. Leppert [1,5,6], Marvin E. Langston[2] & Benjamin I. Chung[1]

Exposure to ambient air pollution has significant adverse health effects; however, whether air pollution is associated with urological cancer is largely unknown. We conduct a systematic review and meta-analysis with epidemiological studies, showing that a 5 μg/m³ increase in $PM_{2.5}$ exposure is associated with a 6%, 7%, and 9%, increased risk of overall urological, bladder, and kidney cancer, respectively; and a 10 μg/m³ increase in $NO_2$ is linked to a 3%, 4%, and 4% higher risk of overall urological, bladder, and prostate cancer, respectively. Were these associations to reflect causal relationships, lowering $PM_{2.5}$ levels to 5.8 μg/m³ could reduce the age-standardized rate of urological cancer by 1.5 ~ 27/100,000 across the 15 countries with the highest $PM_{2.5}$ level from the top 30 countries with the highest urological cancer burden. Implementing global health policies that can improve air quality could potentially reduce the risk of urologic cancer and alleviate its burden.

The global burden of urologic cancer, especially in aging societies, has led to a substantial impact on public health worldwide[1,2]. Nearly 13% of all cancers are urologic cancers, which primarily include prostate, bladder, kidney, and testicular cancers[1]. According to the World Cancer Research Fund International, prostate cancer is the 2nd most frequent cancer in males, with nearly 1.4 million new cases in 2020[3]. Bladder, kidney, and testicular cancer were ranked as the 10th, 14th, and 20th most common cancers worldwide, with nearly 573,000, 430,000, and 74,500 new cases in 2020[3,4].

Urological cancer development is variably affected by modifiable, behavioural, metabolic, and environmental factors[2,5–8]. Environmental exposures, such as cadmium[9], arsenic[8,9], and air pollution[10], have been suggested as factors associated with the risk of urologic cancer. Given few well-defined modifiable risk factors for some urological cancer, especially prostate cancer[11,12], there is an urgent need to evaluate the modifiable environmental risk factors, such as air pollution, as

potential targets for prevention. In light of emerging evidence suggesting the carcinogenic effects of particulate matter (PM), especially its ability to penetrate into multiple organs by causing endothelial damage in vessels through circulation, there is a growing need to investigate the effects of air pollution such as PM exposure in the development of urological cancer[13–15].

Air pollution is a complex and ubiquitous mixture of gases, liquids, and solid particles. Air pollutants vary in chemical composition, reaction characteristics, emission, environmental persistence, capacity to be transferred long or short distances, and health effects. Many countries have established monitoring networks that typically record levels of regulated pollutants, such as respirable particulate matter ($PM_{10}$), fine particulate matter ($PM_{2.5}$), nitrogen dioxide ($NO_2$), sulfur dioxide ($SO_2$), and ozone ($O_3$)[16]. Long-term exposure to air pollution could be associated with cancer risk. In 2013, the International Agency for Research on Cancer (IARC) identified particulate

[1]Department of Urology, Stanford University Medical Center, Stanford, CA, USA. [2]Department of Epidemiology & Population Health, Stanford University School of Medicine, Stanford, CA, USA. [3]Stanford Cancer Institute, Stanford University School of Medicine, Stanford, CA, USA. [4]Department of Epidemiology and Biostatistics, University of California, San Francisco, San Francisco, CA, USA. [5]Division of Nephrology, Department of Medicine, Stanford University School of Medicine, Stanford, CA, USA. [6]Division of Urology, Veterans Affairs Palo Alto Health Care System, Palo Alto, CA, USA. [7]These authors contributed equally: Jinhui Li, Zhengyi Deng. ✉e-mail: jinhuili@stanford.edu

matter (PM) as a human carcinogen[16], specifically to lung cancer. PM with a diameter ≤10 µm[17,18] can penetrate deep into the lungs and enter the circulation, delivering them to different organs[19]. Components of PM, such as heavy metals[20] and polycyclic aromatic hydrocarbons (PAHs)[21], can also induce mutations and initiate or promote carcinogenic processes. Nitrogen oxide ($NO_x$) and nitrogen dioxide ($NO_2$), markers of traffic and fossil fuel emissions, present potential carcinogenic properties that have not been clearly defined[22,23]. The carcinogenic effects of ozone ($O_3$) and sulfur dioxide ($SO_2$) are also unclear, with limited evidence[24–26].

Despite the growing body of evidence suggesting the harmful impact of air pollution on a range of health conditions, including cancer, research examining the potential link between air pollution exposure and urologic cancer risk is sparse. As more epidemiological studies on this topic have been published in the past three years[27–31,28–30,32], it has become both critical and feasible to recapitulate the evidence. In this study, we thus conduct a systematic review and meta-analysis of epidemiological studies to determine potential associations of air pollution exposures with the risk of individual and overall urological cancer.

## Results

### Characteristics of included studies
A total of 5422 studies were identified in electronic databases (Fig. 1). We excluded 1123 duplicate studies, 4215 studies based on title and abstract screen, and 57 studies based on full-text screen, resulting in 27 remaining studies. We further included 10 studies from screening citations of relevant studies and updated literature search. A total of 21 studies were included in the meta-analysis[27–47], among which 13 were published in 2020 or later, and additional 16 studies were included in the systematic review[48–63].

Among all included studies, 12 were based in Europe, 11 in Asia, 10 in North America, 3 in South America, and 1 in Australia (Table 1). There were 18 cohort studies, 10 case-control studies, and 9 ecological studies. Studies evaluated one or more urological cancer types, including overall urologic cancer ($n = 4$), prostate ($n = 21$), bladder ($n = 21$), kidney ($n = 14$), testicular cancer ($n = 3$), and urothelial cancer ($n = 1$). The mean age of the study population ranged from 39.5 to 84.0 years across studies. The air pollutant concentration ranged 3.1–60.3 µg/m³ for $PM_{2.5}$, 5.2–84.3 µg/m³ for $NO_2$, 2.7–107.1 µg/m³ for $PM_{10}$, 8.7–96.4 µg/m³ for $NO_X$, 59.0–87.0 µg/m³ for $O_3$, and 0.66–3.41 µg/m³ for BC. Twenty-seven studies received a high-quality score (score≥6 for case-control and cohort studies; score≥5 for ecological studies) (Supplementary Table S1).

### Associations between air pollutants and risk of urological cancer
We observed that a 5 µg/m³ increase in $PM_{2.5}$ was significantly associated with 7% increased risk of bladder cancer (RR = 1.07, 95%CI: 1.03,1.11; $I^2 = 15.56\%$; $p_{het} = 0.22$), 9% increased risk of kidney cancer (RR = 1.09, 95%CI: 1.04,1.13; $I^2 = 17.58\%$; $p_{het} = 0.37$), and 6% increased risk of overall urological cancer (RR = 1.06, 95%CI:1.03,1.10; $I^2 = 52.36\%$; $p_{het} < 0.001$) (Fig. 2 and Table 3). We also found a 5% non-significantly increased risk for prostate cancer (RR = 1.05, 95%CI: 0.97,1.13; $I^2 = 80.19\%$; $p_{het} < 0.001$), but not for testicular cancer (RR = 1.11, 95%CI: 0.83,1.49; $I^2 = 90.42\%$; $p_{het} = 0.01$). Among 6 studies included in the systematic literature review only, 3 studies reported a statistically significant positive correlation of $PM_{2.5}$ with the risk of prostate and bladder cancer, respectively[51,55,60]; 1 study from Australia reported a non-significant positive association of $PM_{2.5}$ with bladder cancer[62]; 1 study from Hong Kong reported non-significant negative association of $PM_{2.5}$ with urinary cancer[49]; 1 study from Canada showed no significant association between urinary tract cancer associated with traffic-related PM[63].

From 12 studies of $NO_2$ and urological cancer risk, (Fig. 3 and Table 3), a 10 µg/m³ increase of $NO_2$ was marginally associated with a 4% increased risk of prostate cancer (RR = 1.04, 95%CI: 1.00,1.08; $I^2 = 49.83\%$; $p_{het} = 0.02$), a 4% increased risk of bladder cancer (RR = 1.04, 95%CI: 1.00,1.07; $I^2 = 0.00\%$; $p_{het} = 0.45$), and a 3% increased risk of overall urologic cancer (RR = 1.03, 95%CI: 1.00,1.07; $I^2 = 22.26\%$; $p_{het} = 0.039$), but it was not significantly associated with the risk of kidney cancer (RR = 1.06, 95%CI: 0.98,1.14; $I^2 = 47.04\%$; $p_{het} = 0.06$). No study explored the association between $NO_2$ and testicular cancer risk. 7 studies included in the systematic literature review reported a positive association of $NO_2$ with the risk of prostate or bladder cancer[50,53,54,58,60,62]. 1 study failed to identify the significant association between urinary tract cancer and traffic-related $NO_2$ exposure[63].

Meta-analyses of $NO_X$, BC, and $O_3$ did not show statistically significant associations with individual or overall urological cancer, while $PM_{10}$ was associated with a 14% increased risk of prostate cancer (RR = 1.14, 95%CI: 1.02,1.28)(Supplementary Table S2). However, relatively few studies were included in these analyses. Among studies for systematic review only, two studies reported a positive association between $PM_{10}$ and bladder cancer[54,59], and one study found a positive association of high $PM_{10}$ exposure with kidney, prostate, and urothelial cancer (including renal pelvis, ureter, and bladder cancer)[61]. Additionally, one study reported a positive but non-statistically significant association between BC and bladder cancer[62]; one study found a positive association between $SO_2$ and bladder cancer[54]; one study found a positive association between $NO_X$ and overall urological cancer[52]; one study found that ultrafine particles were associated with higher prostate cancer incidence[56]; while no study observed associations for $O_3$, $SO_X$, or CO.

### Subgroup analyses
Table 2 presents the meta-analysis results for the associations of $PM_{2.5}$ and $NO_2$ with overall urological cancer risk by subgroups. RRs of similar magnitude for the association between $PM_{2.5}$ and urological cancer risk were observed by study design, though only that for cohort studies (RR = 1.07; 95%CI:1.03,1.10; $I^2 = 31.45\%$) was statistically significant, with relatively low heterogeneity. Association estimates for $PM_{2.5}$ were also comparable across regions, except for a higher (and least precise) estimate for studies based in Asia (RR = 1.24; 95%CI: 0.35,4.41). Only the estimate for studies from South America (RR = 1.06; 95%CI:1.01,1.11; $I^2 = 25.94\%$) was statistically significant, with lower levels of heterogeneity observed. In analyses by sex, only males showed a significant association for $PM_{2.5}$ exposure (RR = 1.07; 95%CI: 1.02,1.13), though there were many fewer studies of females. Subgroups defined by outcome, age, and country income level demonstrated consistent results.

For the association between $NO_2$ and overall urological cancer risk, case-control and cohort studies showed comparable association estimates with no statistical significance, though only the latter had lower heterogeneity ($I^2 = 9.53\%$). Though only the association for studies of males was marginally significant (RR = 1.04; 95%CI: 1.00,1.09), the fewer studies of females demonstrated a slightly larger and much less precise association (RR = 1.15; 95%CI: 0.29,4.50). Results for $NO_2$ across subgroups were otherwise comparable.

### Publication bias and sensitivity analyses
Based on funnel plots (Fig. 4, Supplementary Fig. S1) and Egger's test, we did not observe a statistically significant publication bias for $PM_{2.5}$ ($p = 0.06$) or $NO_2$ ($p = 0.21$). The trim and fill method did not change the association (Table 3, Supplementary Fig. S2). All sensitivity analyses show robust results compared to the main analyses (Table 3).

### PAF and public health burden
The PAF for overall urological cancer was estimated to be 5.91% (95%CI: 3.61%, 8.16%) for each 5 µg/m³ decrease of $PM_{2.5}$ concentration and 3.05% (95%CI: 0.15%, 5.50%) for each 10 µg/m³ decrease of $NO_2$ concentration (Table 3). The estimated results showed the annual

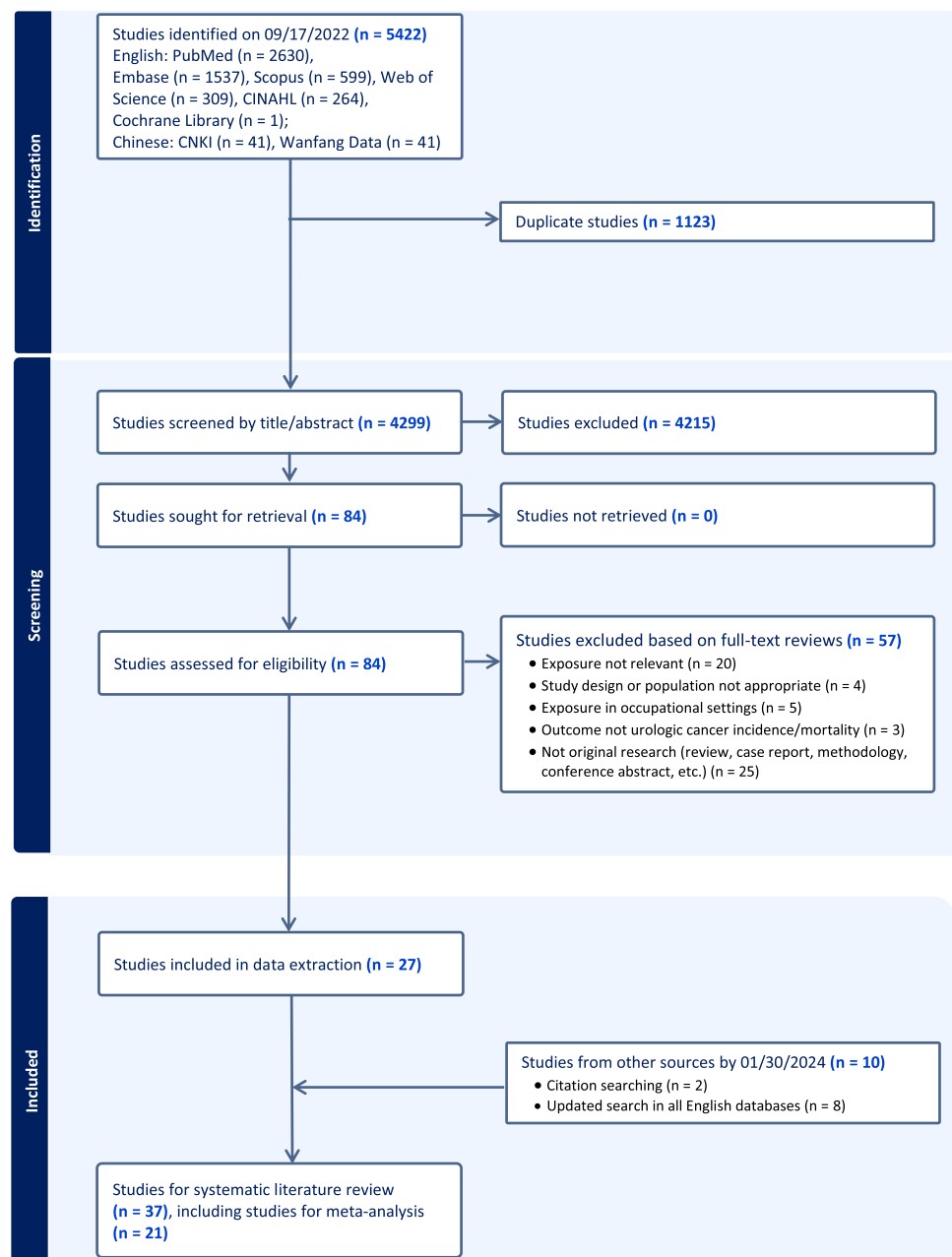

**Fig. 1 | PRISMA flow diagram.** Flow diagram summarises the search strategy and number of studies excluded at each stage. Abbreviations: CINAHL, Cumulative Index to Nursing and Applied Health Literature; CNKI, China National Knowledge Infrastructure.

reduction in ASR and the number of urological cancer cases that could be prevented by reducing the current $PM_{2.5}$ level to 5.8 μg/m³ for the top 30 countries with the highest urological cancer burden, including Egypt, Nigeria, India, China, Iran, etc. (Supplementary Table S3). Figure 5 presents the results for 15 countries with the highest $PM_{2.5}$ levels from these top 30 countries, and shows a reduction in ASR from 1.5 to 27.0/ 100000 across countries.

## Discussion
### Principal findings
To the best of our knowledge, the present study is the first systematic literature review and meta-analysis to comprehensively synthesize associations between multiple air pollutants exposure and the risk of urological cancer. We included 21 epidemiological studies for meta-analysis, including 13 published in 2020 or later in the meta-analysis, from a total of 37 studies in 18 regions/countries for systematic

literature review. Our findings illustrate consistent evidence of an association between higher ambient air pollution exposure and increased urological cancer risk. We identified significantly positive associations between $PM_{2.5}$ and the risk of bladder, kidney, or overall urological cancer, and $NO_2$ with a marginally increased risk of prostate, bladder, and overall urological cancer. Subgroup and sensitivity analyses generally revealed associations that were consistent with the overall analyses. This study provides robust evidence of potential urological cancer risk associated with exposure to air pollution beyond lung cancer.

### Potential mechanisms
It is well known that IARC has identified $PM_{2.5}$ as a leading carcinogen to humans. A recent global review found that chronic exposure can affect every organ in the body, complicating and exacerbating existing health conditions[64]. Nevertheless, whether the associations between

**Table 1 | Contextual details of studies included in the systematic review and meta-analysis**

| Study | | | | | | | Outcome | | | Exposure | | |
|---|---|---|---|---|---|---|---|---|---|---|---|---|
| Study (year) [citation] | Location | Design | Time Period | Number of Participants | Male (%) | Age (average (SD) or range, years) | UCa Type | Outcome | Number of Outcomes | Air Pollutants | Concentration (Average (SD) or range, µg/m³) | Assessment method |
| For systematic review and meta-analysis | | | | | | | | | | | | |
| Felici (2024)[47] | UK | Case-control | 2006~ | Cases: 53,270 Controls: 302,645 | Cases: 46.6 Controls: 46.8 | Cases: 63.76 Controls: 55.63 | PCa BCa KCa | Incidence | PCa: 12,838 BCa: 1516 KCa: 1700 | PM$_{2.5}$ NO$_2$ NO$_x$ PM$_{10}$ | NA | LUR |
| Fan (2023)[29] | China/Jiangsu | Ecological | 2015–2020 | PCa: 43,000,000 KCa/BCa: 84,700,000 | 50.8 | NA | PCa BCa KCa | Mortality | PCa: 13,618 BCa:11,392 KCa: 5,820 | PM$_{2.5}$ | 60.3 (7.0) | Hybrid machine-learning prediction models & |
| Yu (2022) (2)[27] | Brazil | Ecological | 2010–2016 | KCa/BCa: 199,997,499 PCa/TCa:65,496,608 | 48 | NA | PCa BCa KCa TCa | Mortality | PCa: 96,501 BCa:25,019 KCa: 21,018 TCa: 2054 | Wildfire PM$_{2.5}$ | 2.38 (1.62) | CTM |
| Hvidtfeldt (2022)[45] | Europe (Denmark, Sweden, Netherland, France, Austria) | Cohort | 1985–2015 | 302,493 | 0–50 * | 48.2 (13.4) | KCa | Incidence | 847 | PM$_{2.5}$ NO$_2$ O$_3$ BC | PM$_{2.5}$: 15.3 (8.6–19.2) NO$_2$: 24.1 (12.8–39.2) O$_3$: 87.0 (70.3–97.4) BC: 0.88 (0.385–1.155) # | LUR |
| Yu (2022)[28] | Brazil | Ecological | 2010–2018 | KCa/BCa: 147,514,042 | NA | NA | PCa BCa KCa TCa | Mortality | PCa: 127,499 BCa:33,787 KCa: 28,625 TCa: 2802 | PM$_{2.5}$ | 7.63 (3.32) | CTM |
| Youogo (2022)[44] | Canada | Case-control | 1975–1997 | 2844 | 100 | cases: 66.7 (5.6) controls: 65.5 (6.4) | PCa | Incidence | 1420 | PM$_{2.5}$ NO$_2$ | PM$_{2.5}$: 11.9 (3.0) NO$_2$: 29.14 (16.72) | Satellite |
| Taj (2022)[41] | Denmark | Case-control | 1989–2014 | 25,387 | 100 | ≤40 (65%) ^ | TCa | Incidence | 6390 | PM$_{2.5}$ BC NO$_2$ NO$_3$ O$_3$ SO$_2$ SO$_4$ | PM$_{2.5}$: 18.2 BC: 0.85 NO$_2$: 21.75 NO$_3$: 3.71 O$_3$: 58.99 SO$_2$: 13.63 SO$_4$: 3.11 | DEHM/UBM/AirGIS |
| Shin (2022)[30] | South Korea | Cohort | 2005–2015 | PCa: 47,159 BCa: 87,608 KCa: 87,608 | 53.8 | 46.58 (11.01) | PCa BCa KCa | Mortality | PCa: 36 BCa: 27 KCa: 38 | PM$_{2.5}$ PM$_{10}$ | NA | Kriging |
| Huang (2022)[31] | Taiwan | Cohort | 2000–2015 | 189,549 | 100 | 39.5 (12.8) | PCa | Incidence | 732 | PM$_{2.5}$ | 20.81 | Satellite |
| Chen (2022)[32] | Europe (Sweden, Denmark, Netherland, France, Austria) | Cohort | 1985–2015 | 302,493 | 0–50 * | 41.7–72.5 | BCa | Incidence | 967 | PM$_{2.5}$ NO$_2$ BC O$_3$ | PM$_{2.5}$: 14.94 NO$_2$: 24.86 BC: 1.672 O$_3$: 85.44 | LUR |
| Coleman (2020)[34] | USA | Cohort | 1987–2014 | PCa: 282,815 BCa: 635,539 KCa: 635,539 | 44.5 | 45.3 | PCa BCa KCa | Mortality | PCa: 1215 BCa:589 KCa: 603 | PM$_{2.5}$ | 10.7 (2.4) | LUR |

**Table 1 (continued) | Contextual details of studies included in the systematic review and meta-analysis**

| Study (year) [citation] | Location | Design | Time Period | Number of Participants | Male (%) | Age (average (SD) or range, years) | Outcome UCa Type | Outcome Outcome | Number of Outcomes | Exposure Air Pollutants | Concentration (Average (SD) or range, µg/m³) | Assessment method |
|---|---|---|---|---|---|---|---|---|---|---|---|---|
| Coleman (2020) (2)[46] | USA | Ecological | 1992–2016 | 35.4 million [△] | 49.7 | NA | PCa BCa KCa | Incidence | PCa: 1,151,454 BCa: 346,681 KCa: 254,706 | PM$_{2.5}$ | 11.5 (2.6) | LUR |
| Turner (2019)[43] | Spain | Case-control | 1998–2001 | 1911 | Cases: 88 Controls: 87 | cases: 65.8 (9.7) controls: 64.7 (9.8) | BCa | Incidence | 938 | PM$_{2.5}$ NO$_2$ | PM$_{2.5}$: 15.8 (3.89) NO$_2$: 28.6 (10.02) | LUR |
| Shekarrizfard (2018)[40] | Canada/Montreal | Case-control | 2005–2009 | 1722 | 100 | Cases: 65.0 (7.0) | PCa | Incidence | 803 | NO$_2$ | 28.2 | LUR |
| Gandini (2018)[36] | Italy | Cohort | 1999–2008 | 74,989 | 47.3 | 35–65 (70.2%) | BCa KCa | Incidence | BCa: 501 KCa: 196 | PM$_{2.5}$ NO$_2$ | 10-30 (NO$_2$: 76.3%, PM$_{2.5}$: 79.1%) | CTM |
| Pedersen (2018)[37] | Europe (Sweden, Norway, Denmark, Netherlands, England, Austria, Italy, Spain) | Cohort | 1985–~2010 | NO$_2$/NO$_x$: 303,431 Others: 263,634 | 21-55 * | 48 (43–57) * | BCa * | Incidence | NO$_2$/NO$_x$: 943 Others: 827 | PM$_{2.5}$ BC NO$_2$ NO$_x$ PM$_{10}$ PM$_{2.5-10}$ | PM$_{2.5}$: 7.1-30.1 BC: 0.66-3.41 NO$_2$: 5.2-53.2 NO$_x$: 8.7-96.4 PM$_{10}$: 13.5-46.4 PM$_{2.5-10}$: 4.0-16.7 * | LUR |
| Datzmann (2018)[35] | German/Saxony | Cohort | 2007–2014 | 1,918,449 | 46.8 | 49.33 (25.33) | PCa | Incidence | 9611 | PM$_{10}$ NO$_2$ | PM$_{2.5}$: 20.89 NO$_2$: 20.44 | LUR |
| Cohen (2018)[33] | Israel | Cohort | 2004–2015 | BCa: 9,816 PCa:7,509 | 44.7 | 68.2 (12.1) | BCa, PCa | Incidence | BCa: 74 PCa:122 | NO$_x$ | 37.24 | LUR |
| Turner (2017)[42] | USA | Cohort | 1982–2004 | PCa: 278,455 BCa: 623,048 KCa: 623,048 | 40-69 (85%) | PCa BCa KCa | Mortality | | PCa: 1068 BCa:1324 KCa: 927 | PM$_{2.5}$ NO$_2$ O$_3$ | PM$_{2.5}$: 12.6 (2.8) NO$_2$: 21.62 (9.59) O$_3$: 76.4 (8.0) | hybrid LUR and BME |
| Raaschou-Nielsen (2017)[39] | Europe (Sweden, Norway, Denmark, Netherlands, England, Austria, Italy, Spain) | Cohort | 1985–~2010 | NO$_2$/NO$_x$: 289,002 Others: 249,521 | 21-55 * | 48 (43–57) * | KCa | Incidence | NO$_2$/NO$_x$: 697 Others: 603 | PM$_{2.5}$ BC NO$_2$ NO$_x$ PM$_{10}$ PM$_{2.5-10}$ | PM$_{2.5}$: 7.1-30.1 BC: 0.66-3.41 NO$_2$: 5.2-53.2 NO$_x$: 8.7-96.4 PM$_{10}$: 13.5-46.4 PM$_{2.5-10}$: 4.0-16.5 * | LUR |
| Raaschou-Nielsen (2011)[38] | Denmark | Cohort | 1993–2006 | PCa: 25,803 BCa: 53,234 KCa: 46,259 | 47.6 | 56.7 | PCa BCa KCa | Incidence | PCa: 673 BCa:221 KCa: 95 | NO$_x$ | 28.4 | DEHM/UBM/ AirGIS |

For systematic review only [¶]

| Study (year) [citation] | Location | Design | Time Period | Number of Participants | Male (%) | Age (average (SD) or range, years) | Outcome UCa Type | Outcome Outcome | Number of Outcomes | Exposure Air Pollutants | Concentration (Average (SD) or range, µg/m³) | Assessment method |
|---|---|---|---|---|---|---|---|---|---|---|---|---|
| Lim (2023)[62] | Australia | Cohort | 1996–2018 | 11,627 | 100 | 72.1 (4.4) | BCa | Incidence | 224 | PM$_{2.5}$ BC NO$_2$ | PM$_{2.5}$: 5.06 (1.68) BC: 1.07 (0.30) NO$_2$: 13.42 (4.09) | LUR |
| Park (2023)[61] | Korea | Cohort | 2005–2018 | 231,997 | 77.3 | ≥65 (49.5%) | PCa KCa UTCa UCa | Incidence | PCa: 28,440 KCa: 9,736 UTCa: 12,501 UCa: 50,677 | PM$_{10}$ | 56.24 | Monitoring stations |
| Dummer (2023)[63] | Canada | Case-control | 2005–2011 | 1022 | NA | >20 | UCa | Incidence | 219 | NO$_2$ SO$_2$ PM$_{1,0}$ PM$_{2.5}$ | NO$_2$:10.90 (3.95) SO$_2$:0.79 (0.79) PM$_{1,0}$: 2.7 (0.2) PM$_{2.5}$: 3.1 (0.3) | Monitoring stations/LUR |

**Table 1 (continued) | Contextual details of studies included in the systematic review and meta-analysis**

| Study (year) [citation] | Location | Time Period | Number of Participants | Design | Male (%) | Age (average (SD) or range, years) | UCa Type | Outcome | Number of Outcomes | Air Pollutants | Concentration (Average (SD) or range, μg/m³) | Assessment method |
|---|---|---|---|---|---|---|---|---|---|---|---|---|
| Wei (2023)[60] | USA | 2000–2016 | 2,161,156 | Cohort | 100 | 75–84 (88.8%) | PCa | Incidence | 80,615 | NO₂ PM₂.₅ | NO₂: 32.52 (0-239.89) PM₂.₅:9.8 (0–30.9) | GWR† |
| Wang (2019)[55] | China | 2000–2011 | 44.4 million | Ecological | 100 | NA | PCa | Incidence & Mortality | NA | PM₂.₅ | 36–60 (91%) <35 (9%) | Satellite |
| Collarile (2017)@[63] | Italy | 1995–2009 | NA | Ecological | NA | NA | BCa | Incidence | 650 | PM₁₀ NO₂ SO₂ | PM₁₀: 19.6–107.1 NO₂: 10.8–25.5 SO₂: 27.5–85.0 | SPRAY v3 |
| Weichenthal (2017)[57] | Canada/Montreal | 2005–2009 | 2486 | Case-control | 100 | NA | PCa | Incidence | 1240 | ultrafine particles | 24,263/m³ | LUR |
| Cohen (2017)§[52] | Israel | 1992–2013 | 1393 | Cohort | 81 | 54 (8) | UCa | Incidence & Mortality | Incidence:262 Mortality:105 $ | NOₓ | 45.9 (17.2, 160.7) | LUR |
| Yeh (2017)[51] | Taiwan | 2000–2012 | NA | Ecological | NA | NA | BCa | Mortality | NA | PM₂.₅ | NA | Kriging |
| Wong (2016)[49] | Hong Kong | 1998–2001 | 66,820 | Cohort | 35 | ≥65 | UCa | Mortality | 155 | PM₂.₅ | PM₂.₅: 33.7 (3.2) | Satellite |
| Ancona (2015)@[48] | Italy/Rome | 2001–2010 | 85,559 | Cohort | 48.4 | 5–106 | KCa BCa | Incidence & Mortality | KCa: 164 (I), 54 (M) BCa: 477 (I), 73 (M) | SOₓ PM₁₀ | SOₓ:1.67 PM₁₀: 2*10⁻⁵ | SPRAY v5 |
| Shekarrizfard (2015)[56] | Canada/Montreal | 2005–2008 | 1722 | Case-control | 100 | Cases: 65.0 (7.0) | PCa | Incidence | 803 | NO₂ NOₓ | NO₂: 14.87 NOₓ: 788.84 g | LUR |
| Parent (2013)[58] | Canada/Montreal | 2005–2008 | 1772 | Case-control | 100 | Cases: 65.0 (7.0) | PCa | Incidence | 803 | NO₂ | controls: 22.20 (5.08) cases: 22.75 (5.25) | LUR |
| Al-Ahmadi (2013)[50] | Saudi Arabia | 1998–2004 | NA | Ecological | NA | NA | BCa PCa | Incidence | NA | NO₂ | NA | Satellite |
| Yanagi (2012)[59] | Brazil | 1997–2005 | NA | Ecological | NA | NA | BCa | Incidence & Mortality | NA | PM₁₀ | NA | Monitoring stations |
| Liu (2009)[54] | Taiwan | 1995–2005 | 1360 | Case-control | Cases/controls: 67.8 | 50–69 | BCa | Mortality | 680 | PM₁₀ NO₂ O₃ CO SO₂ | PM₁₀: ≤90.29 NO₂: ≤84.32 O₃: ≤71.4 CO: ≤3.42 SO₂: ≤46.82 | Monitoring stations |

†This model integrates ground measurement data, satellite remote sensing products, and atmospheric reanalysis data.

*Range across cohorts.

*median (5–95% percentile).

^65% of participants in the study were younger than 40 years old.

ΔBased on the Surveillance, Epidemiology, and End Results (SEER) website (https://seer.cancer.gov/registries/data.html), the SEER 12 data covers roughly 12.2% of the US population. In 2016, the US population was 323.1 million, and in 1992, the US population was 256.9 million. Here, the average population between 1992-2016 is applied.

¶Included in the systematic review, but did not provide association estimates that could be included in the meta-analysis (i.e., spatial analysis, combined estimates for various cancer types with no specific estimates by UCa type, air pollution from special pollution sources).

@The environmental air pollutants included the source of nearby incinerators or coal-fired and oil-thermal power plants.

§The study population focused on survivors of myocardial infarction.

$These are numbers of all cancer types.

ʸGeographically weighted regressions that ensembled predictions from random forests, gradient boosting, and neural network

BCa bladder cancer, BME Bayesian maximum entropy interpolation model, CTM chemical transport model, DEHM Danish Eulerian Hemispheric Model, KCa kidney cancer, LUR land use regression, NA not available, PCa prostate cancer, SD standard deviation, TCa testicular cancer, UTCa urothelial cancer, UCa urological cancer, UBM Urban Background Model, GWR geographically weighted regression.

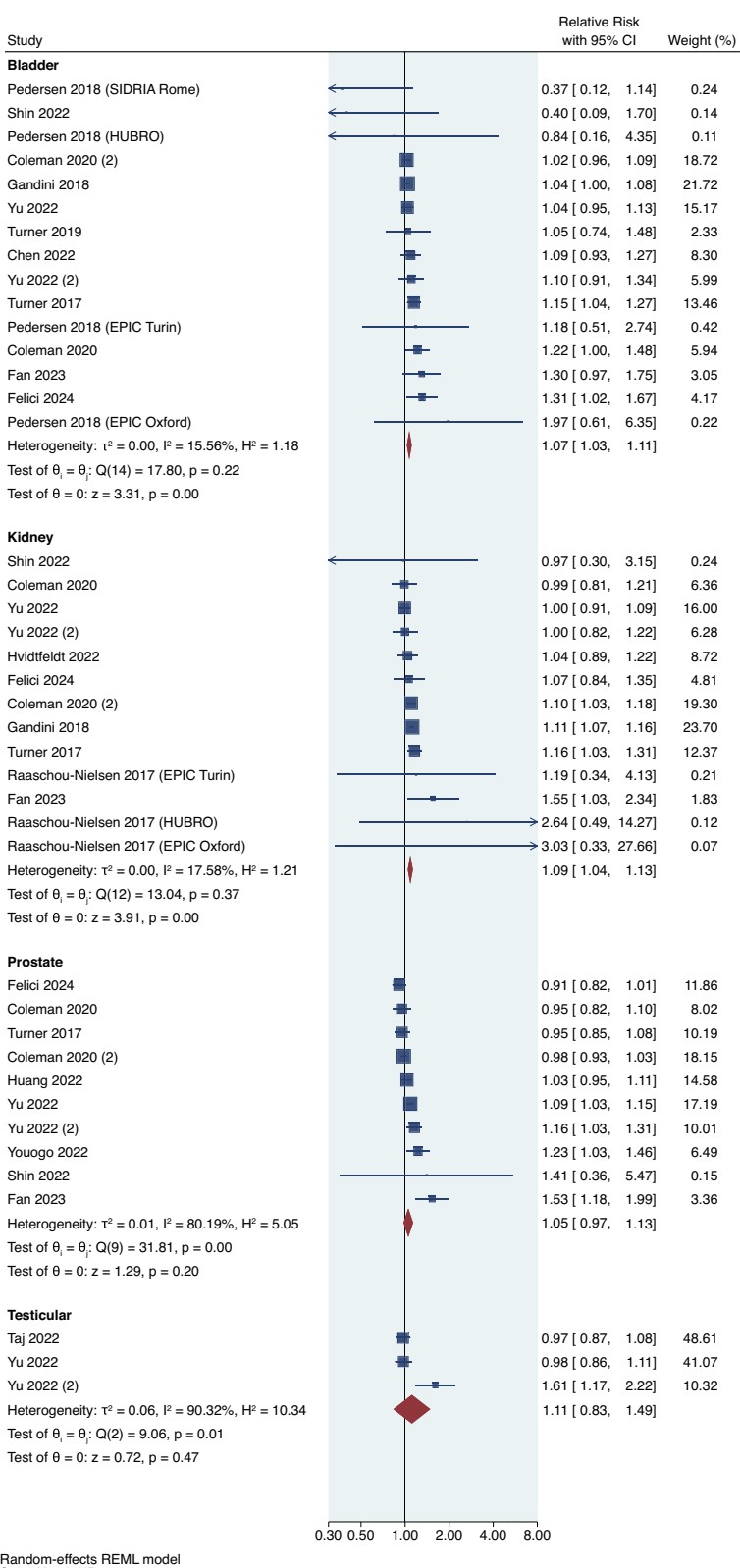

**Fig. 2 | Forest plot of studies reporting PM$_{2.5}$ exposure and urological cancer risk.** Meta-analysis of evidence on the association between a 5 μg/m$^3$ increase in PM$_{2.5}$ and risk of individual urological cancers using random effects meta-analysis. The square represents the relative risk and the bar represents the 95% confidence interval (CI) from each study (n = 41 association estimates which are independent for each cancer type). All statistical tests are two-sided.

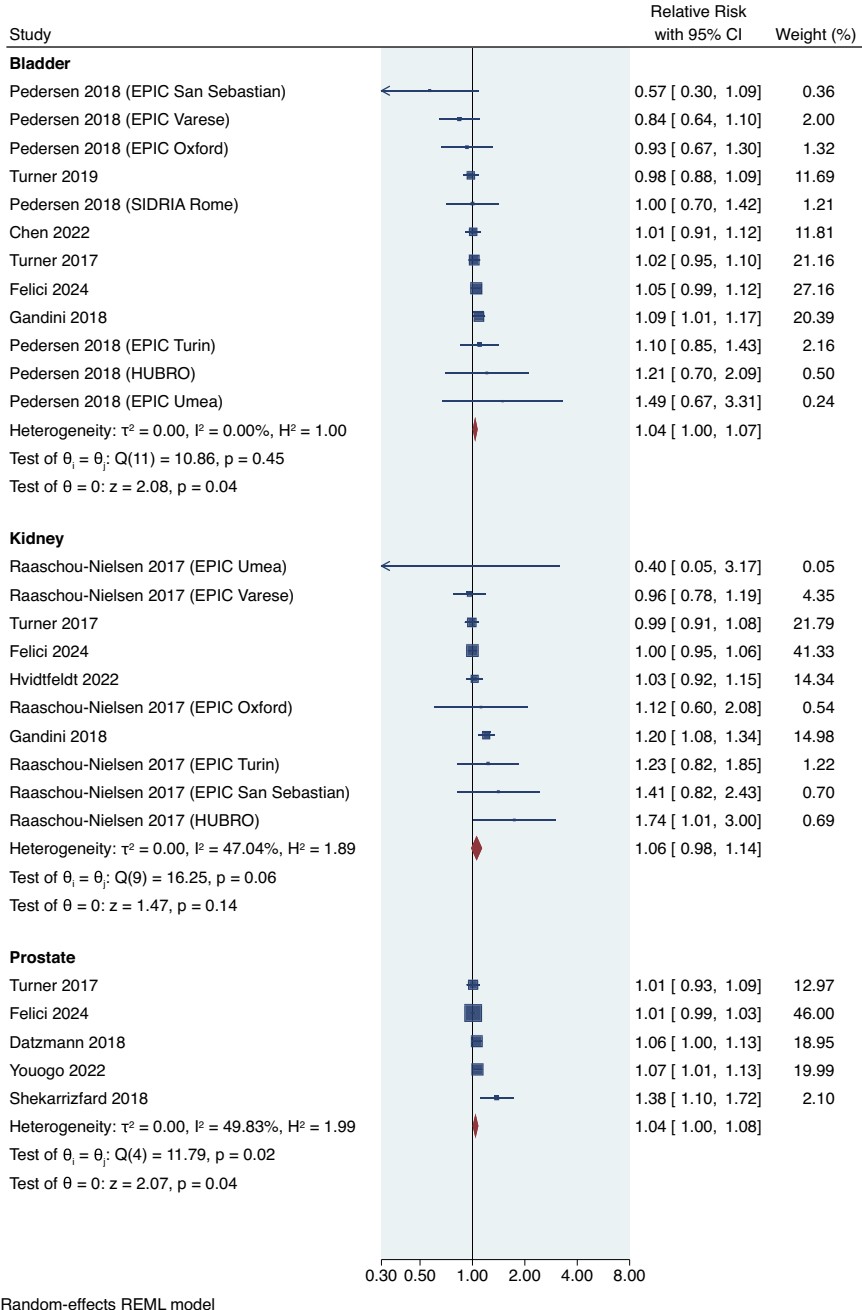

**Fig. 3 | Forest plot of studies reporting NO₂ exposure and urological cancer risk.** Meta-analysis of evidence on the association between a 10 µg/m³ increase in NO₂ and risk of individual urological cancers using random effects meta-analysis. The square represents the relative risk and the bar represents the 95% confidence interval (CI) from each study (n = 28 association estimates which are independent for each cancer type). All statistical tests are two-sided.

PM₂.₅ and urological cancer imply causation and the mechanisms through which PM₂.₅ should affect urological carcinogenesis have yet to be fully understood. PM and its different components are active in a number of processes that contribute to the development of human tumours by promoting the acquisition of biological capabilities required for cancer progression. For example, cellular exposure to PM may activate pathways oriented to "protecting" the integrity of cellular processes, such as activation of P53, RB, and other tumour suppressor genes, which have been considered "gatekeepers" in cancers[65]. Additionally, PM exposure during tumorigenesis has shown harmful effects on cell viability, cellular energetics, and induced immune cell destruction[65]. In rat models, different sizes of PM have been associated with the deregulation of 44 proteins related to energy metabolism and

mitochondrial activity that actively contribute to the metabolic plasticity of cancer cells[66,67]. PM can also generate reactive oxygen species (ROS) in blood, which can induce inflammatory reactions that cause DNA damage[68] and evasion of immune cell destruction[65]. Moreover, PM might contain carcinogens and toxic substances, such as polycyclic aromatic hydrocarbons (PAHs), metals, dioxins, and sulfur-containing compounds that enable the induction of urological cancer[32,69]. The particulate size of PM is considered as another contributor, as smaller particles can reach multiple organs, through circulating system, and thus cause damages to promote cancer development. Miler et al. identified that fine particulate matter in human and animal urine 24 hours or 3 months after exposure, suggesting that kidney played a significant role in PM clearance[14]. Besides, it is well acknowledged that

**Table 2 | Subgroup random-effects meta-analysis with robust variance estimation for associations of a 5 µg/m³ increase in PM$_{2.5}$ and a 10 µg/m³ increase in NO$_2$ with urological cancer risk**

| Pollutant | Category | Study Characteristics (Number of association estimates) | Summary RR | 95%CI | I² |
|---|---|---|---|---|---|
| PM$_{2.5}$ | Study Design | Case-control (6) | 1.06 | 0.87, 1.33 | 65.70 |
| | | Cohort Study (21) | 1.07 | 1.03, 1.10 | 31.45 |
| | | Ecological Study (14) | 1.07 | 0.87, 1.33 | 62.10 |
| | Region | North America (10) | 1.06 | 0.97,1.16 | 65.42 |
| | | Europe (16) | 1.05 | 0.97, 1.12 | 51.40 |
| | | Asia (6) | 1.24 | 0.35, 4.41 | 53.13 |
| | | South America (8) ¶# | 1.06 | 1.01, 1.11 | 25.94 |
| | Outcome | Mortality (20) | 1.09 | 0.97, 1.22 | 53.09 |
| | | Incidence (21) | 1.05 | 1.00, 1.09 | 53.09 |
| | Age | Age≤55 years (12) | 1.02 | 0.97, 1.07 | 0.00 |
| | | Others (29) $ | 1.08 | 1.03, 1.12 | 64.04 |
| | Sex* | Males (25) | 1.07 | 1.02, 1.13 | 69.68 |
| | | Females (12) | 1.04 | 0.88, 1.22 | 41.73 |
| | Income Level& | High (27) | 1.06 | 1.02, 1.09 | 49.79 |
| | | Low/Middle (14) | 1.07 | 0.87, 1.33 | 62.10 |
| NO$_2$ | Study Design | Case-control (7) | 1.02 | 0.97, 1.07 | 19.96 |
| | | Cohort Study (21) | 1.05 | 0.97, 1.13 | 9.53 |
| | Region | North America (5) | 1.04 | 0.75, 1.44 | 7.86 |
| | | Europe (23) | 1.03 | 0.98, 1.08 | 26.78 |
| | Outcome | Mortality (3)^# | 1.01 | 0.97, 1.06 | 0.00 |
| | | Incidence (25) | 1.04 | 0.99, 1.09 | 34.40 |
| | Age | Age≤55 years (9) | 1.03 | 0.93, 1.14 | 14.77 |
| | | Others (19) $ | 1.03 | 0.98, 1.09 | 29.28 |
| | Sex* | Males (9) | 1.04 | 1.00, 1.09 | 48.32 |
| | | Females (3) | 1.15 | 0.29, 4.50 | 64.35 |
| | Income Level& | High (28) | 1.03 | 1.00, 1.07 | 22.26 |
| | | Low/Middle (0) | -- | -- | -- |

Notes:

*Male urological cancers include bladder, kidney, prostate, and testicular cancer; female urological cancers include bladder and kidney cancer.

¶All studies from South America were from Brazil.

#Meta-analysis without robust variance was performed as robust variance could not be estimated from 1 cluster.

$Others include studies that did not report age of study population and studies with population older than 55 years.

^All from the same study Turner 2017.

&The income level was based on the World Bank Statistics.

Abbreviations: CI, confidence interval; RR, relative risk; PM$_{2.5}$, fine inhalable particles, with diameters that are generally 2.5 micrometers and smaller.

PM can impact the cardiorespiratory system by causing endothelial damage in vessels across several organs[13]. Thus, as a high-blood flow organ, the susceptibility of kidney to air pollutant exposure might from PM-related vascular injury[15]. Currently, it is still unclear how long-term exposure to other air pollutants, such as NO$_2$, may contribute to the development of cancer. Some evidence suggests that DNA adduct formation and damage may play a role[70]. Outdoor air pollution is associated with abnormal epigenetic changes, such as DNA methylation, that can modify cancer-related pathways[71,72]. The cumulative biological changes triggered by air pollution exposure over a long time period are likely to contribute to a multistage urological carcinogenesis process involving tumour initiation, promotion, and progression[73]. To thoroughly comprehend the plausible mechanisms of carcinogenesis associated with long-term exposure to PM, NO$_2$, and other gaseous air pollutants, additional research is required from basic science to population-level studies.

## Comparison with other studies

Two prior meta-analyses that focused on air pollution and non-lung cancer incidence and mortality identified only one or two studies focusing on kidney, bladder, or prostate cancer and, therefore, failed to provide conclusive associations[74,75]. Additionally, two recent literature reviews explored the association between air pollutant exposure and urological cancer risk[10,15]. The narrative review from Kim and colleagues[15] focused on the association between PM exposure and urological diseases. Based on the 2 studies on kidney cancer, 6 studies on bladder cancer, and 4 studies on prostate cancer, they reported an inconclusive association of PM with these cancers. Another systematic review from Sakhvidi et al[10]. suggested positive but non-significant associations between specific air pollutants or proxies (e.g., traffic density, proximity index) and bladder, kidney, and urinary tract cancer risk. Unlike our meta-analysis, this review included studies that lacked details of exposure levels and those focused on proxies of industry- or traffic-related air pollution[76–79].

Nearly half of the studies included in our meta-analysis were conducted in Europe[32,35–39,41,43,45,47] and North America[34,40,42,44,46,60,63], where PM$_{2.5}$ levels were relatively low (Europe: 7.1–30.1 µg/m³; USA/Canada: 3.1–12.61 µg/m³). However, a study in American old adults still found that 10-year exposure to PM$_{2.5}$ (mean: 9.8 µg/m³) and NO$_2$ (mean:17.3 µg/m³) was associated with increased risks of prostate cancer[60]. There have been few studies in areas with high air pollution levels, such as Asia, South America, and Africa. One study in Jiangsu, China, reported an annual average concentration of 60.3 µg/m³ for PM$_{2.5}$, and one study in Seoul, Korea, reported an annual average concentration of 48 µg/m³ for PM$_{2.5}$[30]. These levels were over nine times the WHO guideline of an annual mean PM$_{2.5}$ concentration of

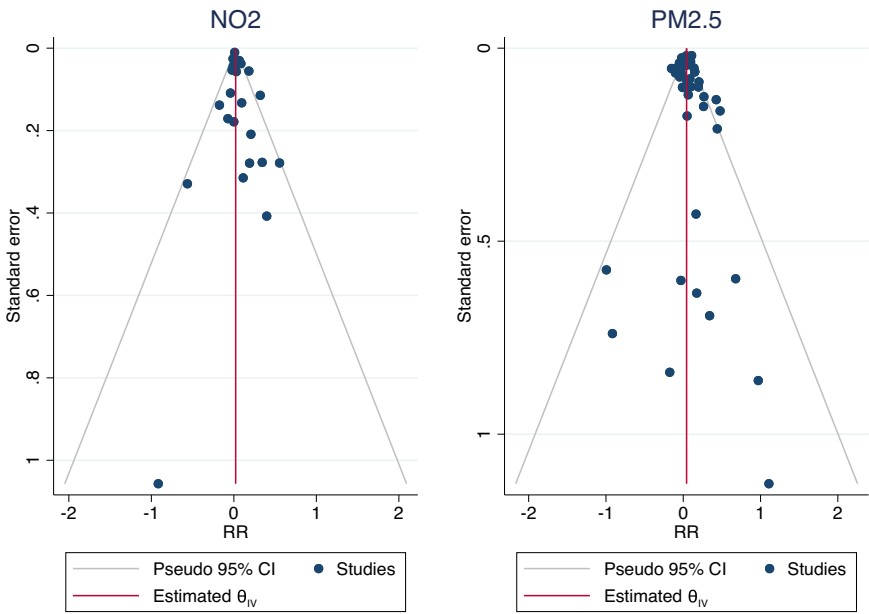

**Fig. 4 | Funnel plots to assess publication bias.** Publication bias in the pooled associations of (left) NO₂ and (right) PM₂.₅ air pollution with overall urological cancer risk.

**Table 3 | Random-effects meta-analysis with robust variance estimation for associations of a 5 µg/m³ increase in PM₂.₅ and 10 µg/m³ increase in NO₂ with urological cancer risk: main analyses, sensitivity analyses (SA), and population attributable fractions (PAF)**

| Pollutant | Meta-analysis | n association estimates | RR (95%CI) | I²(%) | Heterogeneity p-value ^ |
|---|---|---|---|---|---|
| PM₂.₅ | Main analysis uncorrected for publication bias | 41 | 1.06 (1.03, 1.10) | 52.36 | <0.001 |
| | Main analysis corrected for publication bias¶ | 46 | 1.06 (1.02, 1.09) | 51.88 | <0.001 |
| | Sensitivity analyses (SA) | | | | |
| | SA.1 Leave-one-out meta-analysis$ | 40 | 1.07 (1.04, 1.10) | 44.70 | <0.001 |
| | SA.2 Restricted to populations with smoking adjustment | 25 | 1.06 (1.02, 1.10) | 44.43 | 0.012 |
| | SA.3 Restricted to quality assessment score ≥6 | 30 | 1.05 (1.02, 1.08) | 52.26 | 0.002 |
| | SA.4 Restricted to studies with exposure assessment based on LUR modelling | 22 | 1.05 (1.01, 1.09) | 47.89 | 0.014 |
| | SA.5 Restricted to studies published in 2020 or later | 28 | 1.06 (1.01, 1.11) | 52.13 | <0.001 |
| | PAF, % (95%CI)* | | | | |
| | k = 100% | 41 | 5.91 (3.61, 8.16) | -- | -- |
| NO₂ | Main analysis uncorrected for publication bias | 28 | 1.03 (1.00, 1.07) | 22.26 | 0.039 |
| | Main analysis corrected for publication bias¶ | 30 | 1.03 (1.01, 1.05) | 20.51 | 0.026 |
| | Sensitivity analyses (SA) | | | | |
| | SA.1 Leave-one-out meta-analysis$ | 27 | 1.02 (1.00,1.05) | 8.46 | 0.168 |
| | SA.2 Restricted to populations with smoking adjustment | 23 | 1.05 (0.98, 1.12) | 19.81 | 0.070 |
| | SA.3 Restricted to quality assessment score ≥6 | 28 | 1.03 (1.00, 1.07) | 22.26 | 0.039 |
| | SA 4. Restricted to studies with exposure assessment based on LUR modelling | 24 | 1.02 (0.98,1.05) | 0.06 | 0.316 |
| | SA.5 Restricted to studies published in 2020 or later | 7 | 1.02 (0.98,1.05) | 0.00 | 0.448 |
| | PAF, % (95%CI) * | | | | |
| | k = 100% | 28 | 3.05 (0.51, 5.50) | -- | -- |

Notes:

¶ Estimates are from trim-and-fill analysis without robust variance.

$ Estimates are from the meta-analysis that excluded the study that contributed most to heterogeneity (PM₂.₅: Taj 2022 testicular cancer, NO₂: Gandini 2018 kidney cancer) by leave-one-out meta-analyses.

*PAF quantified the proportion of all urologic cancers that are attributable to a 5 µg/m³ increase in PM₂.₅ or a 10 µg/m³ increase in NO₂. We assumed the prevalence of air pollution k = 100% and PAF = (RR-1)/RR. 95%CI was calculated by bootstrap method.

^ All statistical tests are two-sided.

*CI* confidence interval, *KCa* kidney cancer, *NO₂* nitrogen dioxide, *RR* relative risk, *PM₂.₅* fine inhalable particles, with diameters that are generally 2.5 micrometers and smaller.

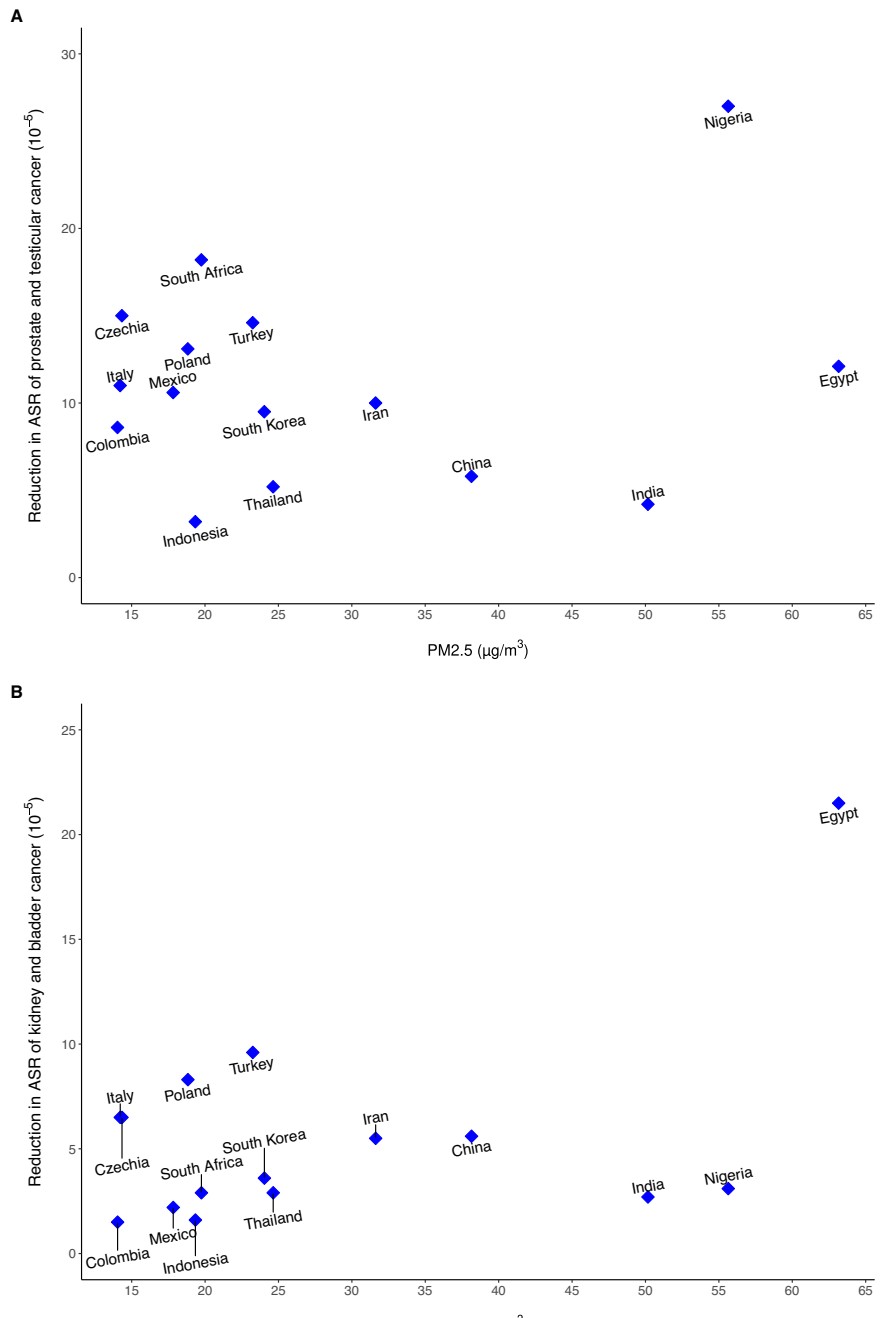

**Fig. 5 | Reduction in urological cancer burden from decreased PM$_{2.5}$ exposure globally.** Annual average PM$_{2.5}$ levels (X-axis) and estimated impact of a reduction in PM$_{2.5}$ to a target level (5.8 μg/m$^3$, below which it is challenging to predict the harmful health effects of PM$_{2.5}$) on age-standard rate (ASR) of individual urological cancer (Y-axis) for top 15 countries with the highest PM$_{2.5}$ level from 30 countries with highest urological cancer burden. A. Reduction in ASR of prostate and testicular cancer; B. Reduction in ASR of kidney and bladder cancer.

5.0 μg/m$^3$ [80]. Two studies from Brazil that used an ecological study design with the annual average concentration of PM$_{2.5}$ and wildfire-related PM$_{2.5}$ as 7.63 μg/m$^3$ and 2.38 μg/m$^3$ [27,28], supported a positive association between PM$_{2.5}$ exposure and prostate cancer risk, but the ecological fallacy is a major concern, and future studies using a prospective cohort study design are needed. More generally, additional studies should be prioritized in developing countries where air pollution levels are higher, and lowering exposure levels would be expected to yield greater public health benefits. This was evident in our analysis of the public health burden attributable to PM$_{2.5}$ among the top 30 countries with the highest urological cancer burden. For example, 75,952 urological cancer cases in China could have been prevented if

the air pollution level could have been reduced to 5.8 μg/m$^3$, under the assumption that the influence of PM$_{2.5}$ was causal. In addition, the correlation between the high incidence of bladder/renal cancer and high PM$_{2.5}$ level in Egypt was noticed. Although Egypt has a high incidence of schistosomiasis-related bladder cancer in history, the successful control of schistosomiasis in Egypt has achieved a substantial decline in the prevalence of schistosomiasis from almost 40% in 1980 to about 1% in 2006 [81]. Accompanied is the remarkable decrease in bladder cancer incidence [82]. Although, schistosomiasis remained as an important risk factor for bladder cancer in Egypt, other emerging etiologic factors, including detrimental air pollution exposure, might also contribute to the high incidence of bladder cancer in this area. We

found a reduction of 12.1 per 100,000 population in the ASR of bladder/kidney cancer, if its current PM$_{2.5}$ level could be reduced to 5.8 µg/m$^3$.

We applied subgroup analyses to explore heterogeneity among the included studies. We observed statistically significant associations and relatively lower heterogeneity in cohort studies for associations of PM$_{2.5}$ with overall urological cancer. Compared to case-control and ecological studies, cohort studies often provide the most robust results due to the prospective collection of individual-level information. We observed a slightly stronger and statistically significant association for PM$_{2.5}$ exposure in males than females. For NO$_2$, females had a relatively stronger association, although it was not statistically significant. It is unclear whether males are more sensitive to PM$_{2.5}$ than females, but a large US cohort study indicated that males had higher all-cause mortality associated with PM$_{2.5}$ exposure[83]. Another study from Japan reported a stronger association between air pollution and CVD emergency care in males than in females[84]. However, other studies contradict these conclusions, demonstrating that females are more susceptible than males to the effects of air pollution[85,86]. It is possible that men have more relative adipose mass, which gives them a larger distribution volume for chemical particles in the environment; or that sex steroid hormones are partially responsible for the differences between males and females[87]. Future studies may consider providing estimates separately for males and females for non-sex-specific cancers, and more sex-specific estimates would still be warranted to resolve sources of heterogeneity.

### Strengths and limitations of the study

This is the first comprehensive meta-analysis of the current epidemiological evidence on ambient air pollution and the risk of urological cancer—made possible by 13 publications since 2020. We evaluated numerous modifiable air pollutants across individual and overall urological cancer. We also conducted the meta-analysis using a novel robust variance estimate that considered the correlation between studies from the same population and provided more valid variance estimates[88].

Several limitations should also be considered. First, several included studies were ecologic in design, with no individual-level data, though the analysis restricted to cohort studies showed similar results. Moreover, given the lack of personal level exposure measurements, there is likely measurement error of ambient pollutants across studies, but we expect this to be non-differential biasing results towards the null. The included studies did not consider the location of participants (outdoors, at home, or at work), and social economic status (SES), and assumed no movement/migration of individuals over the study period. Studies with improved exposure assessment methods, such as portable/personal air monitors, are needed to further clarify the health effects of air pollution. Second, our findings were estimated based on observational studies, where unmeasured and residual confounding from factors such as occupation, passive smoking, and socioeconomic status might bias results. However, the studies included in our meta-analysis considered many potential confounding factors, particularly the most recent publications, and sensitivity analyses restricted to studies with adjustment for smoking status yielded robust results. Third, this study identified a remarkable lack of evidence on the association between air pollution and rare types of urologic cancer, such as cancer in ureter, urethra, and penile. Park et al. found that a high concentration of PM$_{10}$ (≥56 µg/m$^3$) was associated with a 3% increased risk of urothelial cancer, combining cancer in the renal pelvis, ureter, and bladder[61]. More studies are needed to investigate these rare urological cancer types separately. Finally, it is possible that our single-pollutant model could not evaluate possible interaction effects between air pollutants. Future studies should implement mixture models to investigate the interactions of concurrent exposure to multiple air pollutants and time-microenvironment-activity patterns.

### Implications for researchers, clinicians, and policymakers

The ubiquity of ambient air pollution presents a significant public health challenge worldwide, as it has numerous adverse effects on human health, including a possible increased risk of urological cancer. We observed that a 5 µg/m$^3$ reduction in PM$_{2.5}$ concentration and a 10 µg/m$^3$ reduction in NO$_2$ concentration could potentially prevent up to 6% and 3% of urological cancer cases, respectively. These findings imply that air pollution interventions may lessen the personal, public health, economic, and social burden of urological cancer. Currently, the US Environmental Protection Agency (EPA) has updated the primary standards for PM$_{2.5}$ to 9.0 µg/m$^3$ for PM$_{2.5}$[89]. Initiatives to avoid increased exposure to PM$_{2.5}$ may include enacting and enforcing air pollution rules, policies, and laws, transitioning to renewable energy, and maximizing public transit. Our findings also suggest the utility of routine physical examinations and preventative advice for high-risk populations with increased air pollution exposure. Further research that gathers individual-level and precise exposures, long-term follow-up, different groups of susceptible populations, and detailed covariate data is necessary to refine our understanding of appropriate levels of air pollution, dose-response relationships, latency periods, and relevant etiologic time windows toward paving the way for a more comprehensive understanding of the association between air pollution exposure and urological cancer risk.

This meta-analysis emphasizes the need to consider urological cancer as a potential outcome when evaluating exposure to air pollution in public health. The study underlines the potential significance of reducing PM and other air pollutants for mitigating the risk of urological cancer. Moreover, the findings call for high-quality studies investigating the associations between exposure to pollutants and urological cancer risk in middle-/lower-income regions and countries. Overall, our study provides up-to-date evidence on the deleterious effect of air pollution on urological cancer risk and suggests the need for appropriate actions by policymakers and public health authorities to ameliorate this pressing global health issue.

## Methods
### Literature search

The protocol was registered under PROSPERO (CRD42023405773) on 18 March 2023. The study was performed in accordance with PRISMA guidelines[90] (Fig. 1). We searched for all epidemiological studies reporting estimates of associations between ambient air pollution exposure (i.e., air pollution, particulate matter, particles, PM$_{2.5}$, PM$_{10}$, PM$_{2.5-10}$, black smoke, black carbon, NOx, NO$_2$, SO$_2$, CO, and/or O$_3$ and individual or overall urological cancer (i.e., kidney cancer, bladder cancer, prostate cancer, and/or testicular cancer, ureter cancer, urethra cancer, and/or penile cancer) risk. We included literature published by May 11, 2023 that was indexed in PubMed, Web of Science, EMBASE, Cumulative Index to Nursing and Applied Health Literature (CINAHL), Scopus, Cochrane Library, Wanfang Med Online, and China National Knowledge Infrastructure (CNKI). The literature search did not exclude articles based on language or publication date. The search terms for each database were comprehensively verified by the Literature Search Service provided by the Stanford Lane Medical Library (https://lane.stanford.edu/using-lib/lit-search-service.html). Further eligible studies were retrieved by searching the reference lists of relevant narrative and systematic reviews, and an updated search in all English databases (January 30th, 2024). The details of the search strategy are available in Supplementary Appendix 1.

### Selection criteria

Figure 1 illustrates the study selection procedures. COVIDENCE web-based software was applied to assist in collaboration and management of study screening. After removing duplicates, two authors (JL & ZD) independently performed preliminary screening by reviewing the titles and abstracts of the retrieved articles. For articles that passed

preliminary screening, they then performed full-text review to determine eligibility and recorded reasons for exclusion. A senior author (MEL) was recruited for arbitration when discrepancies were encountered. We included studies in the systematic review and meta-analysis that met the following search criteria: 1) epidemiologic study evaluating the association between air pollution and at least one type or all urological cancer risk; 2) cohort, case-control, or ecological study design (the ecological studies were included since air pollution levels are not likely to vary substantially over studied geographic distances); 3) air pollution exposure(s) of $PM_{2.5}$, $PM_{2.5-10}$, $PM_{10}$, $NO_2$, $NO_X$, $O_3$, CO, black carbon (BC, also named $PM_{absorbance}$), and/or $SO_2$. 4) urological cancer outcome(s) such as prostate, bladder, kidney, and testicular cancer. Studies were excluded for the following reasons: 1) no relevant air pollution exposure; 2) no relevant urological cancer outcome; 3) no risk estimate; 4) specialized population (i.e., not adult, occupational-related exposure, participants with specific diseases); 5) conference abstract, letter, animal experiment, clinical trial research study, case report, or review. Concerning multiple publications with overlapping study populations, the meta-analysis included the publication with the most up-to-date estimates, and the others were considered only for context in the systematic review. Additionally, relevant original research did not provide suitable associations for the meta-analysis (i.e., spatial analysis, air pollution from special pollution sources, results for categorical air pollution level only, no relative risk estimates (e.g., absolute risk difference), combined estimates for various cancer types with no specific estimate for urologic cancer type), were included only in the systematic review.

## Data extraction
Data were abstracted in parallel by two authors (JL & ZD), and discordance was solved by a third author (MEL). We contacted the original study authors for additional data or clarification where needed. The following information from each eligible study was abstracted: 1) Citation details (first author, publication year, study period); 2) Study design details (location, sample size, mean age or age range, sex distribution, type of study design); 3) Exposure details (mean levels or range of air pollutants, units of increment); 4) Outcome details (individual/overall urologic cancer), association estimates with 95% confidence intervals (CIs), outcome types (incidence vs. mortality) and the number of cases; 5) Adjustment covariables (e.g., age, sex, smoking, occupation, comorbidities).

## Quality assessment
Two reviewers (JL & ZD) independently used the nine-point Newcastle-Ottawa Quality Assessment Scale (NOS) to assess the quality of case-control and cohort studies, for meta-analysis[91]. A modified NOS with a six-point system was applied for the ecological studies (Supplementary, Table S2). The scale is comprised of three segments: 1) the quality of study selection; 2) the generalizability of the study; 3) the validation of urologic cancer outcome. A star rating system was adopted to assess the quality of the included studies, with each item except for the comparability item being awarded up to one star. For the comparability item, studies were given one star for adjustment for a minimum required set of covariates defined a priori (age, sex, and smoking), and two stars for adjusting additional covariates. For the method of exposure ascertainment, studies that utilized methods beyond air monitors for air pollution concentration, such as the land use regression model (LUR), were considered to have a high-quality exposure assessment. We used a score of ≥6 to define high quality for cohort and case-control studies[10] and a more rigorous score of ≥5 for ecological studies, which were not based on individual exposure.

## Data synthesis and analysis
To investigate the relationship between each air pollutant and urological cancer overall and by cancer types, we assumed a linear relationship and pooled the relative risks (RRs) and 95% confidence intervals (CI) for the following standardized increment of pollutant concentrations determined based on prior literature[92,93]: 5 μg/m³ for $PM_{2.5}$, 10 μg/m³ for $PM_{10}$, $NO_2$, and $NO_X$, and 1 μg/m³ for BC. Estimates were converted from ppb or $10^{-5}$/m to μg/m³ for the needed conversions[94–96]: 1 ppb $NO_2$ = 1.88 μg/m³; 1 ppb $NO_x$ = 1.9125 μg/m³; 1 ppb $O_3$ = 2.0 μg/m³; $10^{-5}$/m BC ($PM_{absorbance}$) = 1.1 μg/m³. Given the rare disease assumption for urological cancer, odds ratios from case-control studies approximated risk ratios. Together with hazard ratios, incidence rate ratios, and risk ratios, they were summarized by meta-analysis to obtain pooled RRs[97]. In addition, we mainly focused on $PM_{2.5}$ and $NO_2$, as the number of studies on other air pollutants was limited (n ≤ 3 for individual urologic cancer type)[98]. For each pollutant, we calculated the pooled RRs by the study-specific estimates using a random-effects model, which is the most conservative approach in this setting as it incorporates within- and between- study heterogeneity in the CI[92]. Two studies on air pollution and kidney cancer conducted pooling projects in multiple cohorts from Europe, where the study populations overlapped. As such, we included the pooled estimates from the most recent study and the estimates for each non-overlapped cohort from the older study[37,39]. The same strategy was applied to two studies on bladder cancer. We also applied robust variance estimation with dependent effect sizes to deal with the effect size multiplicity for any potential overlap populations in the meta-analyses on the analysis for overall urologic cancer estimation[88]. The $I^2$ (supplemented by $\tau^2$ and $H^2$) statistic and Cochrane's heterogeneity Q test were utilized to determine the percentage of variation in effect sizes that could be attributed to between-study heterogeneity[99]. To explore the possible source of heterogeneity, we conducted stratified analyses by study design (case-control, cohort, ecological), geographical location (Asia, North America, Europe, South America), age (≤55, others (not specified or >55)), outcome (incidence, mortality), sex (male, female), and country income level (high, low/middle).

To determine the robustness of our results, we conducted a leave-one-out meta-analysis. Publication bias was also evaluated using funnel plots and Egger's tests for small-study effects[100]. Trim-and-fill analysis with random effects was further applied to estimate the potential effect of unpublished or missing studies on the overall estimates. We conducted sensitivity analyses by restricting to studies with 1) smoking adjustment; 2) quality assessment score ≥6 for case-control and cohort studies; ≥5 for ecological studies (i.e., high-quality studies); 3) exposure assessment based on the Land Use Regression (LUR) model that over half of the included studies applied; and 4) publication year in 2020 or later.

To measure the public health burden of urological cancer attributed to air $PM_{2.5}$ and $NO_2$, we calculated the population-attributable fractions (PAF). To do this, we assumed associations quantified in the meta-analyses reflected causation, and that 100% of the population was exposed to air pollution. We estimated the PAF by (RR-1) / [1 + (RR-1)] and the 95%CI by bootstrap method[93]. Last, $PM_{2.5}$ was used to illustrate the potential impact of reducing air pollution concentration on urologic cancer's public health burden worldwide. We used the World Health Organization's (WHO) estimated urologic cancer cases for each country[101] and the annual average $PM_{2.5}$ concentration (the latest available was in 2019) from WHO[102]. World cancer burden data includes information for 36 cancer types by sex and age group for 85 countries or territories based on the most recent data available to the International Association of Cancer Registries through collaborations with population-based cancer registries, or through information from publicly available databases. WHO collects air pollution data through a combination of independent on-site measurements and data provided by member countries. For each of the 30 countries with highest urological cancer burden, we estimated the annual reduction in age-standard rate (ASR) and

absolute number of urological cancer cases for a reduction of $PM_{2.5}$ concentration from the current annual level to $5.8\,\mu g/m^3$, below which it is challenging to predict the harmful health effects of $PM_{2.5}$[93,103].

We performed analyses using Stata software (version 17; 2023, StataCorp, TX, USA) and R version 4.2.3 (R Foundation for Statistical Computing). The statistical tests were two-sided, and $p < 0.05$ was considered statistically significant.

**Reporting summary**

Further information on research design is available in the Nature Portfolio Reporting Summary linked to this article.

## Data availability

The data used in this study have been deposited in the Figshare database: https://doi.org/10.6084/m9.figshare.25560489.

## Code availability

Stata and R codes are available in the Figshare database: https://doi.org/10.6084/m9.figshare.25560489.

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

## Acknowledgements
We are very thankful for the dedicated efforts of Ms. Connie Wong, the librarian at Stanford University, for her invaluable assistance with the literature search necessary for this meta-analysis.

## Author contributions
JL and ZD contributed equally to this work. MEL and BIC contributed to overall supervision and equally shared the senior authors. JL conceived the study. JL, ZD, MEL, and BIC designed the study. JL, ZD, and MEL collected the data. JL and ZD analyzed the data, drafted, and revised the manuscript. All authors (JL, ZD, SJC, LK, AC, REG, JTL, MEL, BIC) contributed to the data interpretation and critical revision of the intellectual content. All authors (JL, ZD, SJC, LK, AC, REG, JTL, MEL, BIC) have made important intellectual contributions and have seen and approved the final version of the manuscript for submission.

## Competing interests
The authors declare no competing interests.

## Additional information

**Peer review information** *Nature Communications* thanks Mark Linch, Shengzhi Su, and the other, anonymous, reviewer(s) for their contribufion to the peer review of this work. A peer review file is available.

