## [Peer Review File · Nature Communications]

REVIEWER COMMENTS

Reviewer #1 (Remarks to the Author):

Li et al have performed a meta-analysis correlating ambient air pollution to the risk of developing a urothelial cancer. This work is an extension of 2 pan-cancer meta-analyses with the same objectives, but which where urological cancer was unrepresented. The methodology seems sound, and while I would value the input of a statistician, the methodological development presented seems well founded and limitations are highlighted in the manuscript. There are interesting a thought provoking findings; namely that PM2.5 exposure is correlated with increased risk of a number of urological cancers and that NO at 10um/m3 is linked to a higher risk of overall urological cancer risk and prostate cancer. There is also an observation that it was only men that demonstrated the statistically significant association between PM2.5 and urological cancer risk, although the study is probably underpowered to detect such an associated in females who were under-represented in this study. I would like to understand more about the bootstrapping that was performed to demonstrate that individual subgroups (like which country) does not account the the correlations seen. For example, Egypt seem to have the highest PM2.5 and incidence of bladder/renal cancer. However Egypt has historically has high levels of schistosomiasis related bladder cancer not found to such as extent elsewhere.

Overall this is a fascinating piece of work which is hypothesis generating. The major deficit is that there is neither completely novel to the cancer field nor is there any mechanistic work to back up the observations in urological cancer. I therefore, do not think this is sufficient for publication in Nature Communications. However, this work would be well received by the urology community and scientists studying the mechanisms of pollution induced carcinogenesis. I would recommend submission to a tumour specific journal.

Reviewer #2 (Remarks to the Author):

This is a very well written manuscript, with clear explanations of the methodology, the rationale, the findings, and the interpretations. The authors should be congratulated for their high quality work.

I have no further comments as strength and limitations have been addressed appropriately and the methodology is very robust with sensitivity analyses as well as detailed quality assessment of the included papers.

Reviewer #3 (Remarks to the Author):

This systematic review and meta-analysis of epidemiological studies examined the association between exposure to ambient air pollution and urological cancer risk. This study is timely and relevant. The authors found that exposure to elevated levels of PM2.5 and NO2 were linked to increased risks of urologic cancers, including bladder, kidney, and prostate cancer. The potential reduction in urological cancer risk through decreased PM2.5 exposure suggests a need for global policies to enhance air quality and mitigate the public health impact of cancer, particularly urological cancers.

Given the limited well-defined modifiable risk factors for certain urologic cancers and a scarcity of comprehensive quantitative studies exploring the association between ambient air pollution exposure and non-lung-cancer incidence/mortality, especially in the field of urological cancer, the manuscript contributes significantly to the existing body of evidence. The exposure assessment and statistical approaches used in this study are appropriate and robust, with logical and consistent findings across alternative statistical models. These findings carry potential biological and policy implications. The manuscript is generally well-prepared, following pre-registered information and PRISMA guidelines. In summary, this is an excellent research paper that merits publication after incorporating the following comments.

1. Method Section: Consider updating the search, as more than half a year has passed since the last search, including the authors' manual search in PubMed. This update is crucial to ensure no new studies have been overlooked.
2. Method Section: Integrating the research into the systematic review section and discussing findings when there is insufficient statistical assessment for the association evaluation in the meta-analysis section. Providing updates on the search results in the databases would enhance readers' understanding.
3. Figure 5B: Regarding the representation of PAF and ASR reduction, the figure becomes challenging for certain country names and the use of the "square" indicator. Suggesting improvements in clarity would enhance comprehensibility, such as addressing the "blue square" between Italy and Czechia.
4. Population Selection: Provide clarification on why the research population was limited to adults. Additionally, while the authors focused on the main types of urological cancers, consider including more information on rare types. This additional information can be incorporated into the Discussion section to explore if ambient air pollution is associated with the risks of these less common types based on existing biological and epidemiological publications.

Dear Reviewers,

We would like to thank you for your time and salient comments. We addressed each specific comment below and have respectfully attached a revised version of the manuscript (the changes are highlighted in yellow).

Reviewer #1:

Li et al have performed a meta-analysis correlating ambient air pollution to the risk of developing a urothelial cancer. This work is an extension of 2 pan-cancer meta-analyses with the same objectives, but which where urological cancer was unrepresented. The methodology seems sound, and while I would value the input of a statistician, the methodological development presented seems well founded and limitations are highlighted in the manuscript. There are interesting a thought provoking findings; namely that PM2.5 exposure is correlated with increased risk of a number of urological cancers and that NO at 10um/m3 is linked to a higher risk of overall urological cancer risk and prostate cancer. There is also an observation that it was only men that demonstrated the statistically significant association between PM2.5 and urological cancer risk, although the study is probably underpowered to detect such an associated in females who were under-represented in this study. I would like to understand more about the bootstrapping that was performed to demonstrate that individual subgroups (like which country) does not account the the correlations seen. For example, Egypt seem to have the highest PM2.5 and incidence of bladder/renal cancer. However Egypt has historically has high levels of schistosomiasis related bladder cancer not found to such as extent elsewhere.

Overall this is a fascinating piece of work which is hypothesis generating. The major deficit is that there is neither completely novel to the cancer field nor is there any mechanistic work to back up the observations in urological cancer. I therefore, do not think this is sufficient for publication in Nature Communications. However, this work would be well received by the urology community and scientists studying the mechanisms of pollution induced carcinogenesis. I would recommend submission to a tumour specific journal.

Response: Thank you for your thoughtful review of the paper. We highly appreciate the valuable comment and have modified the manuscript accordingly.

First, we agree that although there is a slightly stronger and statistically significant association for PM_{2.5} exposure in males than females, the study is probably underpowered to detect such an association in females who were under-represented in this study due to the small sample size. We have discussed this in the Discussion Section (Line 462-464).

Line 462-464: Future studies may consider providing estimates separately for males and females for non-sex-specific cancers, and more sex-specific estimates would still be warranted to resolve sources of heterogeneity.

Regarding the bootstrap method, it is a method we used to quantify the 95% confidence intervals for the population attributable fraction (PAF), which are presented in Table 3.

Second, thanks for the valuable comment on the correlation between high incidence of bladder/renal cancer and high PM_{2.5} level in Egypt. We agree with the reviewer that Egypt has a high incidence of schistosomiasis-related bladder cancer histologically. Schistosomiasis has been endemic in Ancient Egypt for over 500 decades, and it is a strong risk factor for bladder cancer. However, the most updated studies indicated that the incidence of bladder cancer had declined significantly during the last 25 years in Egypt, and this dropping trend is mainly due to the control and aggressive public awareness campaigns of Schistosomiasis (Khaled 2013). The prevalence of schistosomiasis dropped to 6.6% in 1993, then 1.9% in 2002, and 1.2% in 2006 (El Khoby T, Galal N and A 1998, Khaled 2013). This suggests that increased exposure to other risk factors, including air pollution, might play a role in the high bladder cancer incidence in Egypt. We believe our manuscript would benefit from adding a discussion on schistosomiasis when talking about PM_{2.5} exposure and the risk of bladder cancer in Egypt (Ferguson 1911, Zaghoul 2012). Therefore, we have clarified and added additional deeper discussion on this bladder carcinogen for bladder cancer risk in Egypt, in the Discussion Section (Line 435-445) as follows:

Line 435-445: In addition, correlation between high incidence of bladder/renal cancer and high PM_{2.5} level in Egypt was noticed. Although Egypt has a high incidence of schistosomiasis-related bladder cancer in history, the successful control of schistosomiasis in Egypt has achieved a substantial decline in the prevalence of schistosomiasis from almost 40% in 1980 to about 1% in 2006⁹⁶. Accompanied is the remarkable decrease in bladder cancer incidence⁹⁷. Although, schistosomiasis remained as an important risk factor of bladder cancer in Egypt, other emerging etiologic factors, including the detrimental air pollution exposure, might also contribute to the high incidence of bladder cancer in this area. We found a reduction of 12.1 per 100,000 population in the ASR of bladder/kidney cancer, if its current PM_{2.5} level could be reduced to 5.8 µg/m³.

Third, our study collects both English and Chinese databases worldwide to provide valuable insights through comprehensive data synthesis and rigorous methodology. Its significance lies in its implications for public health policy. As we mentioned in the introduction, the etiologies of most urological cancers are largely unknown, and thus efforts are needed to identify potential causes that will lead to effective prevention strategies. Although there was an increasing number of studies on air pollution and urological cancers, the association remained largely unclear. We conducted the first comprehensive systematic review and meta-analysis on various air pollutants and multiple urological cancer types, and provided robust evidence supporting air pollutants, especially PM_{2.5} and NO₂, as potential urological carcinogens. In the part of “Potential mechanisms” from Discussion Section (Line 361-398), we have included potential biological mechanisms through which air pollutants may increase cancer development, including 1) activation of the pathway of biological capabilities required for the tumor progression from the PM and its components; 2) damaging the cell viability, cellular energetics, and inducing the immune cell destruction; 3) deregulation of proteins related to metabolism and mitochondrial activity; 4) inducing the reactive oxygen species (ROS) in blood and inflammation reactions; 5) formation of DNA adduct from NO₂ and other traffic-related air pollutants. Although evidence-based mechanisms are still limited, there has been an increasing interest in the association between air

pollution exposure and urological diseases, including cancer, considering the high vascularity of the urological system. The toxicity of air pollutants and the effect of the particulate size of PM are considered the main factors in the pathophysiological and carcinogenesis pathways for cancer. The susceptibility of the kidney to environmental pollutants might be due to its high blood flow, high concentration of toxic agents, and high metabolic activity of tubular cells (Kim 2017). The most widely acknowledged mechanism through which PM affects the cardiorespiratory system is by damaging the vascular system, such as causing endothelial injuries to vessels in various organs. PM-related vessel injuries can cause hypertension, which is an established risk factor for renal cancer. Another study detected fine particulate matter in both human and animal urine 24 hours or 3 months after exposure, indicating that the kidney is one of the major clearance pathways for PM exposure (Miller et al. 2017). To make it clearer and embark on more studies exploring the underlying mechanisms of air pollution to the urological diseases, we have added more summary of the existing mechanisms in the Sections of Introduction (Line 76-80) and Discussion (Line 381-388) as follows:

Line 76-80: In light of emerging evidence suggesting the carcinogenic effects of particulate matter (PM), especially its ability to penetrate into multiple organs by causing endothelial damage in vessels through circulation, there is a growing need to investigate the effects of air pollution such as PM exposure in the development of urological cancer ¹³⁻¹⁵.

Line 381-388: The particulate size of PM is considered as another contributor, as smaller particles can reach multiple organs, through circulating system, and thus cause damages to promote cancer development. Miler et al. identified that fine particulate matter in human and animal urine 24 hours or 3 months after exposure, suggesting that kidney played a significant role in PM clearance ¹⁴. Besides, it is well acknowledged that PM can impact the cardiorespiratory system by causing endothelial damage in vessels across several organs ¹³. Thus, as a high-blood flow organ, the susceptibility of kidney to air pollutant exposure might from PM-related vascular injury ¹⁵.

We believe our study is a valuable contribution to the field and of significant interest to the readership of *Nature Communications*.

Reviewer #2:

This is a very well written manuscript, with clear explanations of the methodology, the rationale, the findings, and the interpretations. The authors should be congratulated for their high quality work.

I have no further comments as strength and limitations have been addressed appropriately and the methodology is very robust with sensitivity analyses as well as detailed quality assessment of the included papers.

Response: Thank you for your positive feedback on our manuscript. All coauthors appreciated your time to review our research work.

Reviewer #3:

This systematic review and meta-analysis of epidemiological studies examined the association between exposure to ambient air pollution and urological cancer risk. This study is timely and relevant. The authors found that exposure to elevated levels of PM2.5 and NO2 were linked to increased risks of urologic cancers, including bladder, kidney, and prostate cancer. The potential reduction in urological cancer risk through decreased PM2.5 exposure suggests a need for global policies to enhance air quality and mitigate the public health impact of cancer, particularly urological cancers.

Given the limited well-defined modifiable risk factors for certain urologic cancers and a scarcity of comprehensive quantitative studies exploring the association between ambient air pollution exposure and non-lung-cancer incidence/mortality, especially in the field of urological cancer, the manuscript contributes significantly to the existing body of evidence. The exposure assessment and statistical approaches used in this study are appropriate and robust, with logical and consistent findings across alternative statistical models. These findings carry potential biological and policy implications. The manuscript is generally well-prepared, following pre-registered information and PRISMA guidelines. In summary, this is an excellent research paper that merits publication after incorporating the following comments.

Response: Thank you for your thoughtful review of the paper. We highly appreciate the valuable comment and have revised the manuscript accordingly.

- Method Section: Consider updating the search, as more than half a year has passed since the last search, including the authors' manual search in PubMed. This update is crucial to ensure no new studies have been overlooked..

Response:

We thank the reviewer for the constructive suggestion. We have conducted an updated search in all the English databases on 01/30/2024 and identified 5 additional papers (1 for meta-analysis and systematic review and 4 for systematic review only) to be included in this study. Subsequently, we

have meticulously updated all the analyses and results accordingly. Please find all the updated figures and tables in the revised manuscript and all the updated results have been highlighted. In summary, the updated search and analyses did not change our conclusions on the adverse effects of PM_{2.5} and NO₂ on urological cancer risk.

- Method Section: Integrating the research into the systematic review section and discussing findings when there is insufficient statistical assessment for the association evaluation in the meta-analysis section. Providing updates on the search results in the databases would enhance readers' understanding.

Response: Thanks for your comments and for raising the valuable feedback regarding our study. While updating the search results for the Meta-analysis section, we also updated the search in the Section of Systematic Review only in all English databases as of 01/30/2024. Through this updated search, we identified 5 additional papers, among which 4 of them were qualified for the systematic review only but were lack of sufficient data for the meta-analysis. It is interesting to identify a study from Australia (Lim et al. 2023), which strengthens the diverse study regions in this systematic review and meta-analysis. Based on your recommendation, we have updated the summary results of the review-only papers in the **Results Section** to enhance the readers' understanding (Line 274-280; Line 290-293; Line 296-304). Please find the revision as follows:

For PM_{2.5}:

Line 274-280; Among 6 studies included in the systematic literature review only, 3 studies reported a statistically significant positive correlation of PM_{2.5} with the risk of prostate and bladder cancer, respectively ^{66 70 75}; 1 study from Australia reported a non-significant positive association of PM_{2.5} with bladder cancer ⁷⁷; 1 study from Hong Kong reported non-significant negative association of PM_{2.5} with Urinary cancer ⁶⁴; 1 study from Canada showed no significant association between urinary tract cancer associated with traffic-related PM ⁷⁸.

For NO₂:

Line 288-291: 7 studies included in the systematic literature review reported a positive association of NO₂ with the risk of prostate or bladder cancer ^{65 68 69 71 73 75 77}. 1 study failed to identify the significant association between urinary tract cancer from traffic related NO₂ exposure ⁷⁸.

For other pollutants:

Line 296-304: Among studies for systematic review only, two studies reported a positive association between PM₁₀ and bladder cancer ^{69 74}, and one study found a positive association of high PM₁₀ exposure with kidney, prostate, and urothelial cancer (including renal pelvis, ureter, and bladder cancer) ⁷⁶. Additionally, one study reported a positive but non-statistically significant association between BC and bladder cancer ⁷⁷; one study found a positive association between SO₂ and bladder cancer ⁶⁹; one study found a positive association between NO_X and overall urological cancer ⁶⁷; one study found that

ultrafine particles were associated with higher prostate cancer incidence ⁷¹; while no study observed associations for O₃, SO_X, or CO.

- *Figure 5B: Regarding the representation of PAF and ASR reduction, the figure becomes challenging for certain country names and the use of the "square" indicator. Suggesting improvements in clarity would enhance comprehensibility, such as addressing the "blue square" between Italy and Czechia.*

Response: Thank you for your valuable comment.

To make the figure clearer to the readers, we have updated the analyses based on the updated search results and optimized our images in the revised manuscript. Please find the updated **Figure 5B** attached.

Figure 5B:

- *Population Selection: Provide clarification on why the research population was limited to adults. Additionally, while the authors focused on the main types of urological cancers, consider including more information on rare types. This additional information can be incorporated into the Discussion section to explore if ambient air pollution is associated with the risks of these less common types based on existing biological and epidemiological publications.*

Response: Thanks for your constructive comments.

We focused on adults because urological cancers are more common in older adults than in children or adolescents. In addition, the etiology of urological cancer in children or adolescents might be different from that in adults. Based on the American Cancer Society, unlike many cancers in adults, childhood cancers are thought to be less strongly linked to lifestyle or environmental risk factors.

Our literature search did include all urological cancer types, including prostate, kidney, bladder, testicular cancer, penile, ureter, and urethral cancer (Supplementary Appendix 1). However, we did not find any study that focused on penile, ureter, or urethral cancer. In one study included for systematic review only (Park 2023), the authors evaluated the association between PM₁₀ and urothelial cancer combined, which included cancer in the renal pelvis, ureter, and bladder. They found that a high concentration of PM₁₀ ($\geq 56 \mu\text{g}/\text{m}^3$) was associated with an HR of 1.029 (95%CI: 0.993,1.055) for urothelial cancer. For your reference, the summary of other urological cancer types has been described in the Result Section as follows:

Line 278-280: 1 study from Hong Kong reported non-significant negative association of PM_{2.5} with Urinary cancer ⁶⁴; 1 study from Canada showed no significant association between urinary tract cancer associated with traffic-related PM ⁷⁸.

Line 290-291: 1 study failed to identify the significant association between urinary tract cancer from traffic-related NO₂ exposure ⁷⁸.

Line 296-301: Among studies for systematic review only, two studies reported a positive association between PM₁₀ and bladder cancer ^{69,74}, and one study found a positive association of high PM₁₀ exposure with kidney, prostate, and urothelial cancer (including renal pelvis, ureter, and bladder cancer) ⁷⁶. Additionally, one study reported a positive but non-statistically significant association between BC and bladder cancer ⁷⁷; one study found a positive association between SO₂ and bladder cancer ⁶⁹; one study found a positive association between NO_x and overall urological cancer ⁶⁷;

To clarify it clearer to the readers, we added more details in the Discussion Section of the revised manuscript as follows (Line 488-493):

Line 488-493: This study identified a remarkable lack of evidence on the association between air pollution and rare types of urologic cancer, such as cancer in ureter, urethra, and penile. Park et al. found that a high concentration of PM₁₀ ($\geq 56 \mu\text{g}/\text{m}^3$) was associated with a 3% increased risk of urothelial cancer, combining cancer in the renal pelvis, ureter, and bladder ⁷⁶. More studies are needed to investigate these rare urological cancer types separately.

[Reference]

- El Khoby T, Galal N & F. A (1998) The USAID:Government of Egypt's Schistosomiasis Research Project (SRP). *Parasitol Today*, 14, 92-6.
- Ferguson, A. R. (1911) Associated bilharziasis and primary malignant disease of the urinary bladder with observations on a series of forty cases. *J Pathol Bacteriol*, 16, 76-94.
- Khaled, H. (2013) Schistosomiasis and cancer in egypt: review. *J Adv Res*, 4, 461-6.

- Kim, E. A. (2017) Particulate Matter (Fine Particle) and Urologic Diseases. *Int Neurourol J*, 21, 155-162.
- Lim, E. H., P. Franklin, M. L. Trevenen, M. Nieuwenhuijsen, B. B. Yeap, O. P. Almeida, G. J. Hankey, J. Golledge, C. Etherton-Beer, L. Flicker, S. Robinson & J. Heyworth (2023) Exposure to low-level ambient air pollution and the relationship with lung and bladder cancer in older men, in Perth, Western Australia. *Br J Cancer*, 129, 1500-1509.
- Miller, M. R., J. B. Raftis, J. P. Langrish, S. G. McLean, P. Samutrtai, S. P. Connell, S. Wilson, A. T. Vesey, P. H. B. Fokkens, A. J. F. Boere, P. Krystek, C. J. Campbell, P. W. F. Hadoke, K. Donaldson, F. R. Cassee, D. E. Newby, R. Duffin & N. L. Mills (2017) Inhaled Nanoparticles Accumulate at Sites of Vascular Disease. *ACS Nano*, 11, 4542-4552.
- Zaghloul, M. S. (2012) Bladder cancer and schistosomiasis. *J Egypt Natl Canc Inst*, 24, 151-9.

REVIEWERS' COMMENTS

Reviewer #3 (Remarks to the Author):

The authors have addressed my comments to my satisfaction. This, combined with the positive comments from my fellow reviewers changes my recommendation to suitable for publication in the current form. This is very through provoking work which has been well conducted and will be of interest to a broad church of readers in the field of cancer.

Reviewer #6 (Remarks to the Author):

The authors have addressed my comments adequately.